# Selective Demonstrations for Cross-domain Text-to-SQL

**Shuaichen Chang** and **Eric Fosler-Lussier**
The Ohio State University
{chang.1692, fosler-lussier.1}@osu.edu

## Abstract

Large language models (LLMs) with in-context learning have demonstrated impressive generalization capabilities in the cross-domain text-to-SQL task, without the use of in-domain annotations. However, incorporating in-domain demonstration examples has been found to greatly enhance LLMs' performance. In this paper, we delve into the key factors within in-domain examples that contribute to the improvement and explore whether we can harness these benefits without relying on in-domain annotations. Based on our findings, we propose a demonstration selection framework **ODIS** [1] which utilizes both out-of-domain examples and synthetically generated in-domain examples to construct demonstrations. By retrieving demonstrations from hybrid sources, ODIS leverages the advantages of both, showcasing its effectiveness compared to baseline methods that rely on a single data source. Furthermore, ODIS outperforms state-of-the-art approaches on two cross-domain text-to-SQL datasets, with improvements of 1.1 and 11.8 points in execution accuracy, respectively.

## 1 Introduction

Large language models (LLMs), such as GPT-3 (Brown et al., 2020), Codex (Chen et al., 2021), PaLM (Chowdhery et al., 2022), and LLaMA (Touvron et al., 2023), have demonstrated a great capability of addressing various language tasks with in-context learning (ICL), which rely on prompts that contain a task instruction and zero or a few demonstration examples regarding the task.

Recent studies have evaluated LLMs on the cross-domain text-to-SQL which translates a natural language question (NLQ) to a SQL query for a new database. Previous work has conditioned LLMs with prompts that solely contain the database information without any demonstrations (Rajkumar et al., 2022; Liu et al., 2023) or utilize out-of-domain demonstration examples that are annotated NLQ-SQL pairs associated with the databases that are different from the test database (Poesia et al., 2022; Chen et al., 2023; Pourreza and Rafiei, 2023).

However, Rajkumar et al. (2022) and Chang and Fosler-Lussier (2023) have found that the performance of LLMs can be significantly improved with annotated in-domain examples serving as demonstrations, which are the NLQ-SQL pairs corresponding to the test database. Despite the remarkable performance improvements achieved through in-domain annotated demonstrations, acquiring such data can be costly, as the annotation process requires SQL professionals. More importantly, annotating examples for each new database diminishes the generalizability of text-to-SQL applications. These observations naturally raise two questions: (1) Which are the key factors within the in-domain annotated examples that contribute to the performance improvement? and (2) Can we harness these benefits of in-domain demonstrations without relying on in-domain annotation?

In this paper, we start by investigating the role of the in-domain demonstration examples and then developing new techniques to create demonstrations without leveraging in-domain annotations. We assess three aspects within in-domain annotations: text-to-SQL task knowledge and format, input NLQ distribution, and output SQL distribution. Our experiments show that SQL distribution plays a pivotal role in in-domain demonstrations. This finding motivates us to synthesize in-domain data by generating sampled SQL queries. To the best of our knowledge, we are the first to leverage synthetic examples for text-to-SQL with in-context learning. Furthermore, we introduce a novel demonstration selection framework **ODIS** which utilizes **O**ut-of-domain **D**emonstrations and **I**n-domain **S**ynthetic data. By automatically selecting demonstration

---

```
CreateTable+SelectCol(concert_singer)

-- Using valid SQLite, answer the following
 questions for the tables provided above.
Question: what is the name and nation of the singer
who have a song having 'Hey' in its name?
select name, country from singer where song_name like
'Hey';
Question: How many concerts are there in year 2014 or
2015?
select count(*) from concert where year = 2014 or
year = 2015;
Question: Which year has most number of
concerts?
select
```

Figure 1: An example prompt of 2-shot in-domain text-to-SQL for the database concert_singer in the Spider dataset (Yu et al., 2018). The prompt text of database schema and content, which is constructed using CreateTableSelectCol (Chang and Fosler-Lussier, 2023), is shown at the beginning of the prompt. Two in-domain demonstrations (highlighted in blue) are presented prior to the test question.

```
CreateTable+SelectCol(dorm_1)

-- Using valid SQLite, answer the following
 questions for the tables provided above.
Question: Find the number of students in each major.
select count(*), major from student group by major;
Question: Find the total capacity of all dorms.
select sum(student_capacity) from dorm;

CreateTable+SelectCol(concert_singer)

-- Using valid SQLite, answer the following
 questions for the tables provided above.
Question: Which year has most number of
concerts?
select
```

Figure 2: An example prompt of 2-shot out-of-domain text-to-SQL for the test database concert_singer. An out-of-domain database dorm_1 with 2 associated demonstrations (highlighted in blue) is presented prior to the test database and question.

examples from both out-of-domain and synthetic in-domain data with SQL-guided retrieval methods, our method consistently outperforms both state-of-the-art models employing fine-tuning (Scholak et al., 2021; Li et al., 2023a) and in-context learning (Chen et al., 2021; Pourreza and Rafiei, 2023) on two cross-domain text-to-SQL datasets. Our contributions are in three folds:

- We conduct a thorough analysis to examine the impact of different aspects in in-domain demonstrations for text-to-SQL.
- We propose a demonstration selection framework ODIS that leverages out-of-domain and synthetically generated in-domain demonstrations.
- By employing SQL-guided retrieval methods for selecting out-of-domain and synthetic in-domain demonstrations, ODIS consistently outperforms baselines as well as state-of-the-art approaches.

## 2 Analysis of In-domain Demonstrations

In this section, we analyze the roles and contributions of in-domain annotated demonstrations.

### 2.1 Experiment Setup

We conduct experiments on two widely-used cross-domain text-to-SQL datasets: Spider (Yu et al., 2018) and KaggleDBQA (Lee et al., 2021). For the Spider dataset, we use its development set, which consists of 20 databases with 1047 examples [2]. As for KaggleDBQA, we employ the complete dataset, which contains 8 databases and 272

---

[2] The test set of Spider is not publicly available.

examples. Following Chang and Fosler-Lussier (2023), we utilize a leave-one-out split for evaluating in-domain demonstrations. Specifically, for a given test example, we randomly select demonstrations from the remaining examples associated with the same database, with the condition that selected demonstration examples that do not share the same SQL template as the test example, following the template-split approach (Finegan-Dollak et al., 2018). We also require the selected demonstrations to have different templates following Levy et al. (2022). We repeat all experiments three times to obtain average results. Figure 1 illustrates a prompt example that incorporates two in-domain demonstrations for the database concert_singer in Spider. Detailed prompt examples can be in Appendix A.1. We use Code-davinci-002 version of Codex (Chen et al., 2021) as our LLM due to its demonstrated capability on code generation.

### 2.2 Effectiveness of In-domain Annotations

Figure 3 illustrates the execution accuracy of Codex on Spider and KaggleDBQA. In the zero-shot scenario, where no demonstration examples of NLQ-SQL pairs are provided, Codex achieves execution accuracies of 75.7 and 26.8 for Spider and KaggleDBQA. However, the utilization of in-domain examples demonstrates a considerable improvement in Codex's performance. Notably, as the number of in-domain demonstration examples increases, Codex's performance continues to enhance. With 30 in-domain annotated examples, Codex's performance is significantly boosted to 84.6 and 71.7 on Spider and KaggleDBQA, respectively.

While the benefit of employing in-domain exam-

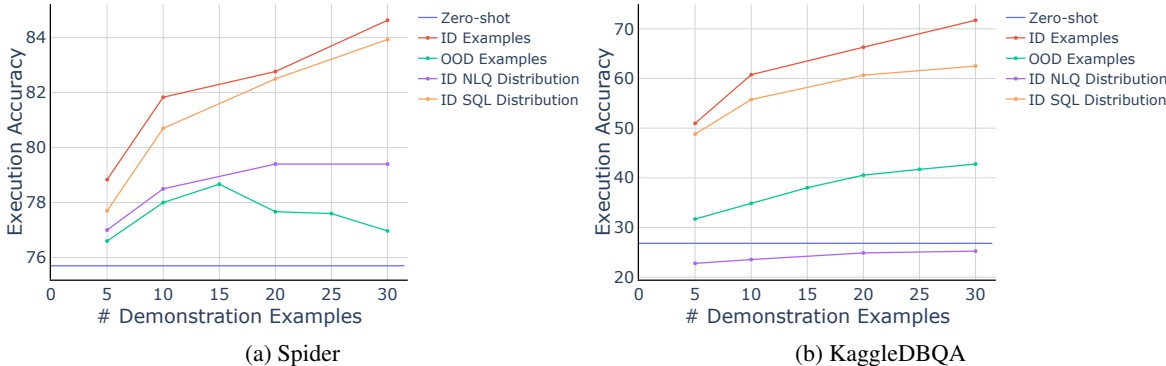

| (a) Spider | (b) KaggleDBQA |
|---|---|

Figure 3: The results of Codex with randomly selected in-domain demonstration examples compared to the zero-shot setting. The x-axis and y-axis represent the number of demonstration examples and the execution accuracy, respectively. ID represents the experiments utilizing in-domain demonstrations while OOD represents the use of out-of-domain demonstrations.

ples in ICL is evident, there still exists a limited understanding of the key factors within in-domain demonstrations that contribute to performance enhancement. To answer this question, we conduct experiments to analyze three aspects of in-domain demonstrations: text-to-SQL task knowledge and format, the distribution of in-domain NLQs, and the distribution of in-domain SQL queries. The first aspect pertains to domain-agnostic knowledge, while the latter two are specific to the domain.

**Text-to-SQL task knowledge and format**   The in-domain examples encompass the task format and knowledge of LLMs, such as the relationship between an NLQ and its corresponding SQL query given a database. To investigate the extent to which the performance gains are due to the task format and knowledge, we conducted a study utilizing annotated out-of-domain data as an alternative source for providing task format and knowledge. In this setup, the model can learn the correct task format and knowledge from the annotated NLQ and SQL pairs associated with other databases. We randomly select examples from the training set of Spider, which contain 140 databases with 7000 examples.

Figure 2 illustrates an example prompt that incorporates out-of-domain demonstrations. We insert $M$ databases preceding the test database, each accompanied by 5 NLQ and SQL pairs. We conducted experiments varying the value of $M$, ranging from 1 to 6. The results of "OOD Examples" shown in Figure 3 demonstrate that exposure to out-of-domain examples with the same task format and knowledge can indeed enhance the model's performance. However, the performance gains from these out-of-domain examples are significantly smaller compared to those obtained from in-domain ex-

amples. Furthermore, we observe that the performance gains achieved through in-domain demonstrations tend to converge or even diminish after a certain number of examples have been provided. This finding stands in contrast to the continuous performance improvement observed with the use of in-domain demonstrations. We believe that while Codex is capable of learning the task format and knowledge from in-domain examples, the task format and knowledge alone are not sufficient to account for the observed performance gains.

In addition, we investigate the necessity of task knowledge in the demonstrations by conducting an experiment where we shuffle the NLQs and SQL queries to create mismatched demonstrations. Figure 7 demonstrates that the performance of Codex is significantly reduced when using demonstrations with mismatched NLQs and SQLs compared to the matched NLQs and SQLs. This finding highlights the necessity of including correct task knowledge and format in the demonstrations. Detailed results of this experiment can be found in Appendix A.2.

**In-domain input NLQ distribution**   Besides task knowledge, in-domain examples also present the input NLQ distribution to LLMs. Recent research has indicated that LLMs can benefit from being aware of the input distribution in classification tasks. Consequently, paring actual inputs with either random output labels (Min et al., 2022) or perturbed outputs (Wang et al., 2022a) has limited impact on the performance of LLMs.

To assess the importance of the input distribution, we substitute the gold SQL queries in the demonstration examples with the predictions generated by Codex in the zero-shot setting. This allows the model to be aware of the distribution of in-

put NLQs without being exposed to the annotated SQLs or any additional knowledge beyond its own predictions. The "ID NLQ Distribution" in Figure 3 indicates that, for Spider, providing in-domain NLQ distribution yields some benefits but it is not comparable to utilizing complete in-domain data. Moreover, for the KaggleDBQA dataset, providing in-domain NLQs with self-predicted SQL queries does not exhibit any advantage compared to not using any examples. We attribute this observation to the lower accuracy of Codex in the zero-shot setting for KaggleDBQA.

**In-domain output SQL distribution** In-domain demonstration examples also serve the purpose of revealing the output SQL distribution to LLMs. To assess the significance of this aspect, we employ the same LLM, Codex, to generate synthetic NLQs from oracle SQL queries. In the demonstration examples, we replace the annotated NLQs with these synthetic NLQs while keeping the SQL queries unchanged. This setup allows the LLMs to be exposed to the in-domain SQL distribution while remaining unaware of the annotated in-domain NLQs or other inputs beyond its own generated NLQs.

The results of "ID SQL Distribution" in Figure 3 demonstrate that providing annotated SQL queries with synthetic NLQs can greatly enhance the model's performance on both the Spider and KaggleDBQA datasets. Furthermore, the performance gains continue to increase with the inclusion of more demonstration examples, aligning with the results obtained using actual in-domain examples.

## 3 Methods

The experimental results presented in Section 2 demonstrate that the SQL distribution encompassed in in-domain annotated examples plays a crucial role in improving performance. However, it is important to note that oracle SQL distributions are not available in most cross-domain text-to-SQL applications. To address this limitation, we propose utilizing synthetic SQL queries and NLQs as in-domain demonstrations to capture the in-domain SQL distribution. As the synthetic examples may not always be correct, relying only on synthetic examples may potentially contradict the task knowledge of text-to-SQL, which could weaken Codex's performance as illustrated in Figure 7. Therefore, to strike a balance between SQL distribution and task knowledge, we leverage out-of-domain data

to provide accurate task knowledge in addition to synthetic data. We propose **ODIS**: a demonstration selection framework that leverages both **O**ut-of-domain **D**emonstrations and **I**n-domain **S**ynthetic demonstrations. In Section 3.1 and 3.2, we present our methods for selecting out-of-domain and in-domain data, respectively, within the ODIS framework.

### 3.1 Out-of-domain Demonstration Creation

Recent work has discovered that LLMs are sensitive to the choice of demonstration examples, and LLMs usually benefit from demonstrations that are similar to the test example (Liu et al., 2021a; Wang et al., 2022b; Rubin et al., 2021). Following that, we further hypothesize that the similarity of the output SQL queries between test examples and demonstration examples is more important than input similarity, as we found that SQL distribution plays a crucial role in the in-domain scenario. Therefore, we propose a retrieval strategy SimSQL to retrieve demonstration examples based on the similarity of predicted SQL queries.

For a test database $d$ and a question $x$, our objective is to retrieve $M$ databases, each with $K$ NLQ and SQL pairs from a set of annotated out-of-domain examples $e_1, ..., e_N$, where $e_i = (db_i, x_i, y_i, \hat{y}_i)$, representing the database, input NLQ, annotated SQL query, and predicted SQL query by an LLM in the zero-shot scenario, respectively. The selected $M$ databases and their associated examples will be presented before the test database and question as shown in Figure 2.

Algorithm 1 illustrates our procedure: (1) We first generate an initial SQL prediction $\hat{y}$ under the zero-shot scenario (line 1); (2) we sort out-of-domain examples based on the similarity between their predicted SQL query $\hat{y}_i$ and $\hat{y}$ (line 2); (3) we scan out-of-domain examples from high similarity to low. Once $K$ examples are found in a database, the database along with these $K$ examples is selected as a demonstration database and examples (line 7-9). The algorithm stops when $M$ databases are selected (line 10-12). We measure the similarity of SQLs with BM25 (Robertson et al., 2009) on the SQL keyword and schema tokens in SQL queries.

### 3.2 In-domain Synthetic Demonstration Creation

In-domain synthetic demonstration selection consists of two stages: (1) synthetic data generation, and (2) synthetic data retrieval.

**Algorithm 1** SimSQL for out-of-domain retrieval

**Input**: a zero-shot text-to-SQL model Model, a test database $d$ and question $x$, and a set of OOD examples $OOD = \{e_1, e_2, ..., e_N\}$ where $e_i = (d_i, x_i, y_i, \hat{y}_i)$, representing the database, input question, gold SQL query, and predicted SQL of Model in the zero-shot setting, respectively.

**Output**: A list of M $Demo[d_i]$, where $Demo[d_i]$ contains $K$ examples associated to $d_i$.

```
1: ŷ ← Model(x)
2: Sort OOD = {eᵢ} in descending order in
   terms of Similarity(ŷ, ŷᵢ)
3: for eᵢ in OOD do
4:     (dᵢ, xᵢ, yᵢ, ŷᵢ) ← eᵢ
5:     if size of Demo[dᵢ] < K then
6:         Demo[dᵢ] ← Demo[dᵢ] + (dᵢ, xᵢ, yᵢ)
7:         if size of Demo[dᵢ] = K then
8:             Output ← Output + Demo[dᵢ]
9:         end if
10:        if size of Output = M then
11:            Break
12:        end if
13:    end if
14: end for
```

**Synthetic data generation**    To generate synthetic data, we follow previous work to first sample synthetic SQL queries $\{y_i\}$ and then translate SQL queries into natural language questions $\{x_i\}$ (Zhong et al., 2020b; Wang et al., 2021; Zhao et al., 2022). We use SHiP (Zhao et al., 2022) to sample synthetic SQL queries, which extract templates from out-of-domain databases and sample columns and values from the test database to fill the templates. After obtaining synthetic SQL queries, we use the Codex to generate corresponding synthetic NLQs, in the same procedure as in our analysis of in-domain SQL distribution.

To improve the quality of synthetic data, we follow the previous work on code generation (Zhong et al., 2020b; Batra et al., 2021) to add a verification process. We use Codex to translate the synthetic NLQ $x_i$ back to SQL $\hat{y}_i$ and filter out the examples that $\hat{y}_i$ and $y_i$ have different execution results.

**Synthetic data retrieval**    While our objective is to synthesize SQL queries that align with the oracle SQL distribution, it is important to note that the oracle SQL distribution often relies on domain-specific prior knowledge that may not be available in the cross-domain text-to-SQL setting. For example, questions commonly asked in a flight booking system may not be easily inferred from those in a concert arrangement system. Therefore, instead of expecting to find SQL queries that closely resemble the test question, our focus is on retrieving multiple queries that cover different aspects of the expected SQL query.

To address this, we formulate the problem as a maximum coverage problem and adopt a greedy algorithm inspired by the algorithm proposed by Levy et al. (2022). Algorithm 2 outlines the process of retrieving demonstrations from synthetic in-domain examples: (1) We begin by creating a set of tokens $S_{uncover}$ that need to be covered, which is initialized with the SQL keyword and schema tokens mentioned in the initial SQL prediction of test question (line 3). (2) We retrieve the synthetic SQLs that have the highest similarity to $S_{uncover}$, measured with BM25 scores (line 5). The $S_{uncover}$ will be updated by removing the tokens in the retrieved SQL query and the retrieved example will be added to the demonstration list (line 8 - 9). We repeat this process until either $S_{uncover}$ becomes empty (line 17) or we are unable to retrieve a synthetic SQL containing any tokens in $S_{uncover}$ (line 14). (3) If the number of selected examples is less than the maximum desired, we iterate through steps (1) and (2) again (line 2).

## 4    Experiments

### 4.1    Baseline Methods

**Previous Work**    We compare ODIS with state-of-the-art approaches that either require finetuning on the Spider training set or utilize in-context learning. For the finetuning-based methods, we select SmBoP (Rubin and Berant, 2021) which utilizes RoBERTa-large as the pretrained model, as well as T5+Picard (Scholak et al., 2021), ShiP+Picard (Zhao et al., 2022), and RESDSQL (Li et al., 2023a), which employ T5-3B (Raffel et al., 2020) as the pretrained model.

For the in-context learning methods, we include the approaches proposed by Rajkumar et al. (2022), Chang and Fosler-Lussier (2023), Lan et al. (2023) which utilize either no demonstrations or randomly selected demonstrations. We also include SYN-CHROMESH (Poesia et al., 2022) and SKILL-KNN (An et al., 2023), which employ similarity-based retrieval methods. Additionally, we select state-of-the-art approaches LEVER (Ni et al., 2023), Self-Debug (Chen et al., 2023), and DIN-

**Algorithm 2** `CovSQL` for in-domain retrieval

**Input**: a zero-shot text-to-SQL model `Model`, a test database $d$ and question $x$, and a set of synthetic in-domain examples $ID = \{e_1, e_2, ..., e_N\}$ where $e_i = (x_i, y_i)$.

**Output**: A list of $K$ examples from $ID$.

```
 1: ŷ ← Model(x)
 2: while size of Output < K do
 3:     S_uncover ← tokens in ŷ
 4:     while size of S_uncover > 0 do
 5:         e* ← argmax Sim(S_uncover, y_i)
                e_i∈ID
 6:         x*, y* ← e*
 7:         if Sim(S_uncover, y*) > 0 then
 8:             S_uncover ← S_uncover − tokens in y*
 9:             Output ← Output + e*
10:             ID ← ID − e*
11:             if size of Output = K then
12:                 Break
13:             end if
14:         else
15:             Break
16:         end if
17:     end while
18: end while
```

SQL (Pourreza and Rafiei, 2023), which incorporate fixed demonstration examples with intermediate reasoning steps or self-correction procedures.

**Ours**   To demonstrate the effectiveness of ODIS framework, we compare it to baselines that utilize demonstrations solely from out-of-domain data or synthetic in-domain sources. Furthermore, we evaluate the proposed SQL-guided retrieval strategies `SimSQL` and `CovSQL` by comparing them with the `Random` retrieval strategy, as well as the `SimNLQ` approach, which retrieves examples based on the similarity of input NLQs. In line with previous studies (Rubin et al., 2021; Shi et al., 2022), we measure similarity using the cosine distance of sentence embeddings encoded through Sentence-BERT (Reimers and Gurevych, 2019).

### 4.2   Experiments Setup

We utilize the training set of Spider as the pool for selecting out-of-domain demonstrations, following our experiments in Section 2. We generate and filter synthetic in-domain examples, resulting in 1416 examples for Spider and 512 examples for KaggleD-BQA. When the `Random` retrieval strategy is used, we conduct three repetitions with different random seeds and report the average results. We conducted experiments with both closed-source LLMs, Codex and ChatGPT, in addition to the open-source LLM, CodeLLama, for the final prediction [3]. Due to resource constraints, we only employ Codex for retrieving demonstrations.

**Hyper-parameters**   For out-of-domain demonstrations, we employ 5 NLQ-SQL pairs for each demonstration database, in line with our experiments in Section 2. Determining the number of out-of-domain databases and the number of synthetic in-domain examples is considered a hyper-parameter selection process. In our final experiments for Spider and KaggleDBQA, we set the number of out-of-domain databases to 4 and synthetic in-domain examples to 5, based on the results obtained from experiments conducted on a randomly-selected subset of 20 databases from the Spider training set.

## 5   Results

### 5.1   Main Results

Table 1 and Table 2 present the execution accuracy of state-of-the-art methods and our proposed ODIS on Spider and KaggleDBQA. On Spider, ODIS with Codex achieves an execution accuracy of 85.2, surpassing both the state-of-the-art finetuning-based model RESDSQL (Li et al., 2023a) and the in-context learning method Self-Debugging (Chen et al., 2023) by 1.1 points. On KaggleDBQA, ODIS achieves an execution accuracy of 54.8, outperforming the state-of-the-art model SKILL-KNN (An et al., 2023) by 11.8 points. ODIS also demonstrates superior performance compared to baselines that solely rely on out-of-domain or synthetic in-domain demonstrations, regardless of the backbone LLM used. On the Spider dataset, ODIS outperforms these baselines by 3.1 and 3.6 points when using Codex, by 0.8 and 3.0 points with ChatGPT, and by 4.4 and 1.7 points with CodeLlama. Likewise, on KaggleDBQA, ODIS surpasses the baselines by 1.5 and 9.6 points when employing Codex, by 7.3 and 19.8 points with ChatGPT, and by 4.4

---

[3] We utilize OpenAI APIs `code-davinci-002` and `gpt-3.5-turbo-16k-0613` for Codex and ChatGPT. We opt for `CodeLlama-34B-instruct` in CodeLlama experiments.

[4] The reported execution accuracy is measured using the Spider official test-suite evaluation (Zhong et al., 2020a), which may yield different values compared to those presented in the original method papers if different evaluation scripts were utilized.

| Method | LLM | EX |
|---|---|---|
| *Previous Work with Finetuning* | | |
| SmBoP (Rubin and Berant, 2021) | RB | 78.0 |
| T5+Picard (Scholak et al., 2021) | T5-3B | 79.1 |
| ShiP+Picard (Zhao et al., 2022) | T5-3B | 81.4 |
| RESDSQL+NatSQL (Li et al., 2023a) | T5-3B | 84.1 |
| *Previous Work with In-context Learning* | | |
| Rajkumar et al. (2022) | Codex | 67.0 |
| Chang and Fosler-Lussier (2023) | Codex | 76.8 |
| LEVER (Ni et al., 2023) | Codex | 81.9 |
| DIN-SQL (Pourreza and Rafiei, 2023) | Codex | 75.6 |
| DIN-SQL (Pourreza and Rafiei, 2023) | GPT-4 | 82.8 |
| Self-Debugging (Chen et al., 2023) | Codex | 84.1 |
| *This work* | | |
| Zero-shot | Codex | 75.7 |
| | ChatGPT | 75.7 |
| | CodeLlama | 70.7 |
| Out-of-domain Demo Only | Codex | 82.1 |
| | ChatGPT | 80.7 |
| | CodeLlama | 75.6 |
| Synthetic In-domain Demo Only | Codex | 81.5 |
| | ChatGPT | 78.5 |
| | CodeLlama | 78.3 |
| ODIS | Codex | **85.2** |
| | ChatGPT | 81.5 |
| | CodeLlama | 80.0 |

Table 1: The execution accuracy (EX) on the Spider development set.[4] The upper section contains models that are finetuned on the Spider training set, while the middle and bottom sections showcase previous methods and our proposed methods, which use in-context learning. The column LLM denotes the language model utilized, either in the fine-tuning or in-context learning. RB represents RoBERTa-large.

and 7.4 points with CodeLlama. These results highlight the effectiveness of leveraging both sources of demonstrations within the ODIS framework.

## 5.2 Analysis

We evaluate our demonstration retrieval methods `SimSQL` and `CovSQL`, in comparison to baseline retrieval methods. To specifically analyze and compare the performance of the retrieval methods within each demonstration source, we evaluate the retrieval methods for out-of-domain demonstrations without utilizing in-domain synthetic demonstrations, and vice versa.

**Out-of-domain retrieval** Table 3 presents the comparison among different retrieval methods when retrieving out-of-domain examples. It is observed that retrieving out-of-domain examples with similar predicted SQL queries outperforms random selection and NLQ similarity-based selection on both datasets. This finding aligns with our earlier observations that the output SQL distribution plays a more crucial role than the input NLQ distribution in improving the model's performance.

| Method | LLM | EX |
|---|---|---|
| *Previous Work with Finetuning* | | |
| SmBoP (Rubin and Berant, 2021) | RB | 27.2 |
| T5+Picard (Scholak et al., 2021) | T5-3B | 29.8 |
| *Previous Work with In-context Learning* | | |
| Zero-shot (Lan et al., 2023) | Codex | 23.9 |
| Few-shot (Lan et al., 2023) | Codex | 40.4 |
| SKILL-KNN (An et al., 2023) | Codex | 43.0 |
| *This work* | | |
| Zero-shot | Codex | 26.8 |
| | ChatGPT | 25.7 |
| | CodeLlama | 18.8 |
| Out-of-domain Demo Only | Codex | 53.3 |
| | ChatGPT | 45.6 |
| | CodeLlama | 37.9 |
| Synthetic In-domain Demo Only | Codex | 45.2 |
| | ChatGPT | 33.1 |
| | CodeLlama | 34.9 |
| ODIS | Codex | **54.8** |
| | ChatGPT | 52.9 |
| | CodeLlama | 42.3 |

Table 2: The execution accuracy (EX) on the KaggleD-BQA.

We also explore the use of oracle SQL queries in our experiments. Retrieving examples with the oracle SQL query of the test question results in an additional improvement of 0.8 and 2.9 points in execution accuracy on the Spider and KaggleDBQA datasets, respectively. We believe that the larger improvement observed on KaggleDBQA can be attributed to the lower performance of Codex on this dataset in the zero-shot setting. This suggests that employing a better initial model may lead to further enhancements in the performance of the ODIS framework.

| Method | Spider | Kaggle |
|---|---|---|
| Random | 78.7 | 38.0 |
| SimNLQ | 81.2 | 42.6 |
| SimSQL | **82.1** | **53.3** |
| SimSQL (Oracle) | 82.9 | 56.2 |

Table 3: The execution accuracy of Codex on Spider and KaggleDBQA with different out-of-domain demonstration retrieval methods.

**Synthetic in-domain retrieval** Table 4 provides a comparison of the in-domain retrieval methods. On the Spider dataset, retrieving synthetic SQL queries that cover different parts of the initial

| Method | Spider | Kaggle |
|---|---|---|
| Random | 77.0 | 38.2 |
| SimNLQ | 79.2 | 41.5 |
| SimSQL | 79.7 | **45.2** |
| CovSQL | **81.5** | **45.2** |
| CovSQL (Oracle) | 81.8 | 48.9 |
| CovSQL (In-domain Annotation) | 82.4 | 52.9 |

Table 4: The execution accuracy of Codex on Spider and KaggleDBQA with different synthetic in-domain demonstration retrieval methods.

| Method | LLM | EX |
|---|---|---|
| *Previous Work with Finetuning* | | |
| SmBoP (Rubin and Berant, 2021) | RB | 58.1 |
| T5+Picard (Scholak et al., 2021) | T5-3B | 65.0 |
| *This work* | | |
| Zero-shot | Codex | 65.3 |
| Out-of-domain Demo Only | Codex | 70.9 |
| Synthetic In-domain Demo Only | Codex | 70.9 |
| ODIS | Codex | **73.4** |

Table 5: The execution accuracy (EX) of Codex on the Dr. Spider NLQ post-perturbation sets.

SQL predictions outperforms the retrieval methods based on the similarity of either NLQs or initial SQL predictions. On the KaggleDBQA dataset, both the `CoverSQL` and `SimSQL` methods achieve the same results, outperforming the random selection and NLQ-similarity-based methods. Regarding the experiments with oracle SQL queries, again, we observe a larger improvement on the KaggleD-BQA dataset compared to Spider which may be attributed to the lower performance of Codex on KaggleDBQA.

**Impact of synthetic data**   To assess the impact of synthetic data quality in ODIS framework, we compare two SQL synthesizing methods SHiP (Zhao et al., 2022) and GAZP (Zhong et al., 2020b). Both methods extract the templates of SQL queries from the Spider training set and fill the template with the database schema and content with a new database, however, SHiP leverages a schema-weighted sampling to make the synthetic SQL queries more realistic by controlling the relationship of sampled columns. When replacing our synthetic data generated through SHiP to GAZP, we find the performance with Codex has dropped from 81.5 to 79.5 on Spider and from 45.2 to 33.8 on KaggleDBQA. This indicates that the quality of synthetic data is crucial in the ODIZ framework.

Additionally, we evaluate the performance of retrieving from synthetic in-domain examples compared to retrieving from actual annotated in-domain examples in Spider and KaggleDBQA with the `CovSQL` retrieval method. The performance gap in Table 4 between them indicates that refining the data synthesizing method could potentially yield even greater enhancements within the ODIS framework. It is worth noting that this experiment is designed to illustrate the limitations of synthetic examples compared to annotated examples. Re-

trieving annotated in-domain examples typically necessitates a large amount of in-domain annotations, which are usually not available.

**Robustness Evaluation**   In order to assess the robustness of our proposed ODIS framework, we conduct a study to compare ODIS with the baseline supervised-learning and in-context-learning methods on Dr. Spider (Chang et al., 2023), an evaluation benchmark for text-to-SQL robustness. We opt for the NLQ post-perturbation sets in Dr. Spider, as these have been identified as the most challenging for LLMs with in-context learning (Chang et al., 2023). The results, presented in Table 5, reveal that ODIS with Codex achieves an execution accuracy of 73.4 on NLQ perturbations, surpasses the supervised learning methods and the Codex with either out-of-domain or synthetic in-domain demonstrations.

## 6   Related Work

**Text-to-SQL with in-context learning**   Previous studies have explored text-to-SQL with in-context learning from various perspectives. Rajkumar et al. (2022) and Chang and Fosler-Lussier (2023) investigated effective approaches to represent databases in prompts. Other studies have explored the incorporation of intermediate reasoning steps to enhance the performance of LLMs (Chen et al., 2023; Pour-reza and Rafiei, 2023; Tai et al., 2023).

For demonstration retrieval, prior work has explored different strategies for retrieving out-of-domain examples based on similarity (Poesia et al., 2022; Rubin et al., 2021) and diversity (Levy et al., 2022; Su et al., 2022). Concurrent with our work, Nan et al. (2023) suggested retrieving out-of-domain demonstrations based on the initial SQL predictions, which aligns with our retrieval method.

Additionally, execution results have been used in the SQL generation process to verify the gen-

erated SQL queries (Chen et al., 2023; Ni et al., 2023; Nan et al., 2023; Guo et al., 2023). These approaches often require multiple LLM calls and SQL executions to accomplish majority voting or self-correction. It is worth noting that the database execution time is a crucial consideration in realistic applications, as highlighted by Li et al. (2023b). In contrast, ODIS with our proposed retrieval strategy only requires two LLM calls (one for initial SQL prediction and one for final generation) and eliminates the need for any database execution, making it a more efficient solution for text-to-SQL tasks.

**Data synthesis for Text-to-SQL** Due to the high cost of acquiring annotated text-to-SQL data, previous research has explored the use of synthetic data as an alternative approach to enhance the performance of cross-domain text-to-SQL models. These works primarily focused on data augmentation, such as generating additional examples for the training databases (Guo et al., 2018; Yu et al., 2021; Wang et al., 2021; Wu et al., 2021; Yang et al., 2021; Liu et al., 2021b; Zhao et al., 2022). Zhong et al. (2020b) first synthesized data for test databases and adapted pretrained text-to-SQL models on the synthetic data associated with the test database before inference. In a similar vein, ODIS leverages synthetic data for the test database but uses it with in-context learning rather than directly fine-tuning models on it.

## 7  Conclusion and Future Work

In this work, we delve into the analysis of crucial aspects of in-domain demonstrations and identify the SQL distribution as the key factor. Inspired by our findings, we propose a novel demonstration selection framework, ODIS, which harnesses the power of both out-of-domain demonstrations and synthetic in-domain examples using SQL-guided retrieval. The remarkable performance across different backbone large language models demonstrates the effectiveness of our proposed framework, compared to both baseline and state-of-the-art methods.

While ODIS serves as a general framework that can be employed with various demonstration retrieval methods, our experiments in this study utilize separate retrieval methods for out-of-domain and in-domain demonstrations. We regard it as future work to explore a unified retrieval strategy that breaks the boundary between out-of-domain and synthetic in-domain data, enabling automatic

selection among them. Additionally, our retrieval approach relies on the predictions of Codex in the zero-shot scenario. It is worth exploring utilizing higher-performance initial text-to-SQL models to further enhance the performance, as demonstrated in Section 5.2 through the use of oracle SQL queries. We believe the effectiveness of the ODIS framework will encourage further advancements in data synthesis methods and demonstration retrieval techniques.

Considering the proven effectiveness of utilizing hybrid data sources, comprising out-of-domain examples and synthetic in-domain examples as in-context demonstrations, we believe that ODIS can be extended to few-shot, parameter-efficient fine-tuning by leveraging these hybrid data sources (Hu et al., 2021; Dettmers et al., 2023). We leave the exploration of parameter-efficient fine-tuning with synthetic in-domain examples for future research.

## Limitations

We hope the effectiveness of ODIS will draw attention to the potential benefits of incorporating hybrid demonstrations – synthetic and annotated data in the text-to-SQL task in in-context learning. As the first study exploring this approach, we examined the retrieval methods separately for these two types of demonstrations. However, as mentioned in Section 7, exploring more advanced retrieval methods to blur the boundary of out-of-domain and synthetic in-domain examples is a direction for future work.

We conducted our experiments with both closed-source Codex [5] and ChatGPT [6] and open-source CodeLlama as the large language models. Recent research has shown that GPT-4 consistently outperforms Codex in the text-to-SQL generation when employing various prompt strategies (Pourreza and Rafiei, 2023). Considering our budget constraints, we leave the ODIS framework with other large language models as a topic for future investigation.

While our proposed method involves two calls to LLMs and no SQL execution during the generation process, it does require an offline process for synthetic example generation. We acknowledge that this process adds an additional step to the pipeline. However, we believe that it can be seamlessly integrated with the database deployment process. By

---

[5]Codex is currently free for research, with API at https://openai.com/api/. For commercial use, Codex is available at https://azure.microsoft.com/.

[6]ChatGPT API is available at https://openai.com/api/.

incorporating the synthetic example generation step during the database deployment phase, users querying the database will not experience any noticeable delays or disruptions.

## Ethics Statement

We acknowledge the importance of the ACL Ethics Policy and agree with it. The objective of text-to-SQL systems is to provide non-expert database users with the ability to query databases using natural language. Our research in this paper focuses on improving the generalization capabilities of text-to-SQL models. While the benefit of in-domain annotated demonstrations is evident, the significant cost associated with such annotation should not be ignored in real-world applications. Therefore, we propose a method that combines both out-of-domain examples and synthetic in-domain data to enhance models' generalization and reduce reliance on costly in-domain annotations. We believe that this approach can improve the performance and usability of text-to-SQL systems, making them more accessible and practical for a wider range of applications.

Moreover, it is important to consider the cost of LLMs and SQL execution in realistic scenarios. Our proposed method employs two calls to LLMs without requiring any SQL execution during the SQL decoding process. This design choice aims to optimize the cost and user experience compared to other approaches that involve multiple LLM calls and SQL executions.

## Acknowledgments

We would like to thank Michael White, Micha Elsner, the OSU SLaTe lab, as well as the anonymous reviewers for their valuable feedback on this work. Shuaichen Chang is supported in part by the Amazon Post Internship Fellowship.

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

# A Appendix

## A.1 Prompt Construction

For the Spider dataset (Yu et al., 2018), we utilize the `CreateTableSelectCol` (Chang and Fosler-Lussier, 2023) prompt construction to create the prompt text for databases. In the KaggleDBQA dataset (Lee et al., 2021), each database is accompanied by a database document that provides descriptions of the columns in the database. We incorporate the database document as comments within the `CreateTableSelectCol` prompt. Figure 4 and 5 provide examples of the database prompt text for Spider and KaggleDBQA, respectively.

## A.2 Necessity of Text-to-SQL Task Knowledge

Figure 6 shows a prompt example with mismatched NLQs and SQL queries as demonstrations. In this example, the first NLQ corresponds to the second SQL and the second NLQ corresponds to the first NLQ. Figure 7 shows that the performance of Codex is significantly reduced when using demonstrations with mismatched NLQs and SQLs compared to the matched NLQs and SQLs. This finding highlights that it is important to maintain the correct task knowledge in the demonstrations, which inspires us to leverage out-of-domain demonstrations along with synthetic in-domain examples.

```
CreateTable+SelectCol(concert_singer)

create table stadium (
stadium_id int,
location text,
name text,
capacity int,
highest int,
lowest int,
average int,
primary key (stadium_id)
);
/*
Columns in stadium and 3 distinct examples in each column:
stadium_id: 1, 2, 3;
location: "Raith Rovers", "Ayr United", "East Fife";
name: "Stark's Park", "Somerset Park", "Bayview Stadium";
capacity: 10104, 11998, 2000;
highest: 4812, 2363, 1980;
lowest: 1294, 1057, 533;
average: 2106, 1477, 864;
*/

create table singer (
singer_id int,
name text,
country text,
song_name text,
song_release_year text,
age int,
is_male bool,
primary key (singer_id)
);
/*
Columns in singer and 3 distinct examples in each column:
singer_id: 1, 2, 3;
name: "Joe Sharp", "Timbaland", "Justin Brown";
country: "Netherlands", "United States", "France";
song_name: "You", "Dangerous", "Hey Oh";
song_release_year: 1992, 2008, 2013;
age: 52, 32, 29;
is_male: "F", "T";
*/

create table concert (
concert_id int,
concert_name text,
theme text,
stadium_id text,
year text,
primary key (concert_id),
foreign key (stadium_id) references stadium(stadium_id)
);
/*
Columns in concert and 3 distinct examples in each column:
concert_id: 1, 2, 3;
concert_name: "Auditions", "Super bootcamp", "Home Visits";
theme: "Free choice", "Free choice 2", "Bleeding Love";
stadium_id: 1, 2, 10;
year: 2014, 2015;
*/

create table singer_in_concert (
concert_id int,
singer_id text,
primary key (concert_id,singer_id),
foreign key (concert_id) references concert(concert_id),
foreign key (singer_id) references singer(singer_id)
);
/*
Columns in singer_in_concert and 3 distinct examples in each column:
concert_id: 1, 2, 3;
singer_id: 2, 3, 5;
*/
```

Figure 4: An database prompt example of for the `concert_singer` database in Spider.

```
CreateTable+SelectCol(StudentMathScore)

create table finrev_fed_17 (
state_code integer, -- the state code of the finrev_fed_17
idcensus integer, -- the idcensus of the finrev_fed_17
school_district text, -- the school district of the finrev_fed_17
nces_id text, -- the nces id of the finrev_fed_17
yr_data integer, -- the year data of the finrev_fed_17
t_fed_rev integer, -- total federal revenue through the state to each school district.
c14 integer, -- federal revenue through the state- title 1 (no child left behind act).
c25 integer -- federal revenue through the state- child nutrition a
);
/*
Columns in finrev_fed_17 and 3 distinct examples in each column:
state_code: 33, 5, 14;
idcensus: 33203100130100, 5501905900000, 14501615800000;
school_district: "NEW YORK CITY SCHOOL DISTRICT", "LOS ANGELES UNIF SCH DIST", "CITY OF
CHICAGO SCHOOL DISTRICT 299";
nces_id: 3620580, 622710, 1709930;
yr_data: 17;
t_fed_rev: 2061297, 1146298, 783943;
c14: 956851, 376182, 290912;
c25: 439209, 390711, 200517;
*/

create table ndecoreexcel_math_grade8 (
year integer, -- the year of the ndecoreexcel_math_grade8
state text, -- the state of the ndecoreexcel_math_grade8
all_students text, -- the all students of the ndecoreexcel_math_grade8
average_scale_score integer -- the average scale score of the ndecoreexcel_math_grade8
);
/*
Columns in ndecoreexcel_math_grade8 and 3 distinct examples in each column:
year: 2017;
state: "National", "Alabama", "Alaska";
all_students: "All students";
average_scale_score: 283, 268, 277;
*/

create table finrev_fed_key_17 (
state_code integer, -- the state code of the finrev_fed_key_17
state text, -- the state of the finrev_fed_key_17
#_records text -- the number of records of the finrev_fed_key_17
);
/*
Columns in finrev_fed_key_17 and 3 distinct examples in each column:
state_code: 1, 2, 3;
state: "Alabama", "Alaska", "Arizona";
#_records: 137, 54, 235;
*/
```

Figure 5: An database prompt example of for the StudentMathScore database in KaggleDBQA.

```
CreateTable+SelectCol(concert_singer)

-- Using valid SQLite, answer the following
  questions for the tables provided above.
Question: what is the name and nation of the singer
who have a song having 'Hey' in its name?
select count(*) from concert where year = 2014 or
year = 2015;
Question: How many concerts are there in year 2014 or
2015?
select name, country from singer where song_name like
'Hey';
Question: Which year has most number of
concerts?
select
```

Figure 6: An example prompt of 2-shot in-domain text-to-SQL using mismatched demonstration examples.

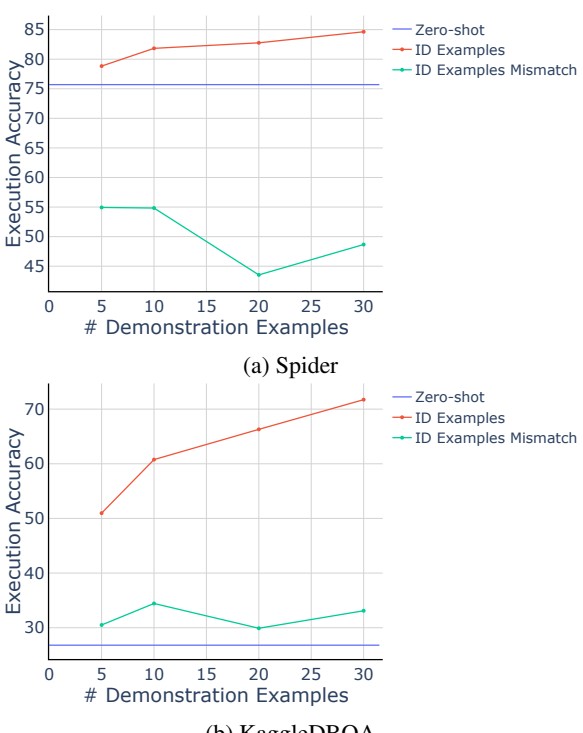

(a) Spider

(b) KaggleDBQA

Figure 7: The results of Codex with real in-domain demonstration examples compared to mismatched examples. `ID Examples` represents annotated in-domain demonstrations while `ID Examples Mismatch` represents the NLQ and SQL pairs are mismatched. The x-axis and y-axis represent the number of demonstration examples and the execution accuracy, respectively.