# OpenReview forum: "Selective Demonstrations for Cross-domain Text-to-SQL"
_EMNLP/2023/Conference — EMNLP 2023 Findings_

### Official Review · Reviewer_E6z7 · 2023-07-28

**Soundness:** 4

**Excitement:**

3: Ambivalent: It has merits (e.g., it reports state-of-the-art results, the idea is nice), but there are key weaknesses (e.g., it describes incremental work), and it can significantly benefit from another round of revision. However, I won't object to accepting it if my co-reviewers champion it.

**Paper Topic And Main Contributions:**

This paper proposes to use both synthetic in-domain and out-of-domain text-to-SQL data to improve the performance of text-to-SQL task for a new domain for large language models with in-context learning.

The paper first examines the importance of two parts in the in-domain demonstration: the text (natural language question) part and the SQL part, and obtains the observation that SQL part contributes more during the in-context learning process.

Based on this finding, they propose to use synthetic in-domain data where the SQL queries are first generated and then translated into natural language questions following previous works. Combined with the out-of-domain demonstrations, they propose a demonstration selection framework ODIS. Both the in-domain and out-of-domain demonstrations are selected based on similarity (or coverage) with the input data.

Experiments show both the out-of-domain demonstration only method and synthetic in-domain demonstration only method outperform the zero-shot setting using Codex on two datasets by 5-6% execution accuracy. ODIS further shows 3% execution accuracy over in-(out-of-) domain only method.



**Questions For The Authors:**

1.	For Weakness 1, is the difference due to different demonstration selection methods? Will the finding hold using the same selection method.
2.	What’s the number of demonstrations used in each method in Table 1 and does the number of demos affect the results? If so, an analysis on this would be valuable.

Minor question: DIN-SQL method shows using GPT-4 outperforms Codex by 7% accuracy. Will applying ODIS to GPT-4 achieve even higher accuracy? Providing a discussion based on a subset is ok if the budget is limited.

**Reasons To Accept:**

1.	The analysis of which part of the demonstration is important is valuable. Although the analysis is only done for the text-to-SQL task, I believe it can be easily extended to other tasks.
2.	The idea of combining synthetic in-domain and real out-of-domain demos during in-context learning is novel and effective, which shows SOTA results for two datasets.


**Reasons To Reject:**

1.	The results in Figure 3 do not match the results in Tables 1 and 2. For example, in Figure 3, out-of-domain only demonstrations show very poor performance (around 78%). However, Table 1 shows 82.1%. And it seems that the proposed method (85.2%) achieves better performance than using the gold in-domain demonstrations (around 84%).

**Reproducibility:**

4: Could mostly reproduce the results, but there may be some variation because of sample variance or minor variations in their interpretation of the protocol or method.

**Reviewer Confidence:**

3: Pretty sure, but there's a chance I missed something. Although I have a good feel for this area in general, I did not carefully check the paper's details, e.g., the math, experimental design, or novelty.

---

> ### Author Rebuttal · Authors · 2023-08-29
>
> We appreciate that the reviewer provided a nice summary of our work and found both of our analysis and proposed method valuable and novel. We also thank the reviewer for their helpful comments and we have made a response to each of their comments.
>
>
> > The results in Figure 3 do not match the results in Tables 1 and 2.
>
> We apologize for the confusion. The results in Figure 3 are achieved with randomly selected demonstrations (both in-domain and out-of-domain). The results in Table 4,5 are achieved with our proposed retrieval methods. In Section 2, we aimed to analyze what aspects of gold in-domain annotations provide the most benefits. As a result, we did not consider complex demonstration retrieval methods. We used randomly selected demonstrations for both in-domain and out-of-domain, and the experiments were repeated 3 times. For
> The results in Section 5 are about our proposed demonstration selection framework OSID, where we used the proposed SimSQL and CovSQL to retrieve examples.
>
>
> > is the difference due to different demonstration selection methods? Will the finding hold using the same selection method.
>
> The difference between the results in Figure 3 and in Tables 1 and 2 is due to the chosen method of demonstration selection, as we clarified in response to the earlier question. Regarding the finding in Figure 3, the experiments in Figure 3 are conducted where the selection method remained the same (random selection in three separate runs). It's also worth noting that when comparing "ID Examples", "ID NLQ Distribution", and "ID SQL Distribution", the exact same examples are used across each run. The only distinction is the aspect of the gold in-domain examples that are presented to the LLM.
>
>
>
>
>
> > What’s the number of demonstrations used in each method in Table 1 and does the number of demos affect the results? If so, an analysis on this would be valuable.
>
> In all our experiments, we consistently maintain the number of examples per out-of-domain database at 5 to reduce the complexity of experiments. For the results in Tables 1 and 2, we configure 4 out-of-domain databases (each with 5 examples) and 5 synthetic in-domain examples for both spider-dev and KaggleDBQA, resulting in the utilization of a total of 25 examples. These numbers are regarded as hyperparameters, which are selected based on the results obtained on a randomly selected subset of 20 databases from the Spider training set.
>
> We observed that the performance is affected by the amount of demonstrations. When increasing the amount of both in-domain demos and out-of-domain databases, we witnessed a pattern wherein Codex's performance initially ascends and subsequently descends.
> We also found that the optimal number of examples is tied to the chosen retrieval method. For example, the optimal number of out-of-domain examples is 15 for random retrieval (as shown in Figure 1a) but the optimal amount of out-of-domain examples is 20 for SimSQL.
>
> Being the pioneering effort to harness a hybrid-source demonstration approach (combining out-of-domain and synthetic in-domain data), our work focuses on demonstrating the effectiveness of the proposed framework. We believe that conducting a comprehensive analysis of the impact of the number of demonstrations is important in the ODIS framework and regard this as future work.
>
> > Minor question: DIN-SQL method shows using GPT-4 outperforms Codex by 7% accuracy. Will applying ODIS to GPT-4 achieve even higher accuracy? Providing a discussion based on a subset is ok if the budget is limited.
>
> We greatly appreciate this invaluable suggestion. We conducted an experiment with GPT-4 in the final prediction stage on a subset of Spider due to our limited budget. Specifically, we focused on the extra-hard examples, as defined in the Spider paper [1], which are recognized as the most challenging examples in terms of SQL structure. This subset, drawn from the Spider development set, contains a total of 166 examples. We believe that employing the most challenging data affords a clearer illustration of our methodology, within the constraints of a modest budget.
> We use identical prompts for the studies with Codex and GPT-4, where the demonstrations were selected based on Codex's initial predictions. We suspect that substituting Codex with GPT-4 for the demonstration retrieval stage could yield a further enhancement for ODIS. However, this transition incurs significant costs, hence rendering it unfeasible at the present juncture.
>
> The table below contains our new experiment with GPT-4 on the Spider extra-hard
> |Method|LLM|Spider extra-hard|
> |---|---|---|
> |ODIS|Codex|65.1|
> |ODIS|GPT-4|68.7|
>
>
>
> Moreover, to demonstrate the effectiveness of our proposed method with other LLMs, we conducted an experiment with ChatGPT (gpt-3.5-turbo-16k), which is more than 10X cheaper than GPT-4, and Code Llama (CodeLlama-34b-Instruct), which achieves the best performance on code-related tasks among open-sourced LLMs, on the entire Spider and KaggleDBQA datasets. The results are presented below. Both the out-of-domain demonstration only method and synthetic in-domain demonstration only method outperform the zero-shot setting on two datasets, with GPT-3.5 and Code Llama. Notably, the ODIS approach shows even more improvements in comparison to both the in-domain only and out-of-domain only methods. Due to our limited budget, we did not undertake a hyperparameter search for the experiments with GPT3.5 and Code Llama. Nevertheless, the current results already illustrate the generalizability of our approach.
>
> As a result, we believe the remarkable performance of ODIS with both closed-source and open-sourced LLMs on two cross-domain text-to-SQL datasets is able to demonstrate the generalizability of our proposed approach.
>
>
> |Method|LLM|Spider|Kaggle|
> |---|---|---|---|
> |Zero-shot|GPT-3.5|75.7 | 25.7|
> |Out-of-domain Only|GPT-3.5| 80.7| 45.6|
> |Synthetic In-domain Only|GPT-3.5|78.5 | 33.1|
> |ODIS|GPT-3.5|**81.5** |**52.9**|
> ||||
> |Zero-shot|Code Llama|70.8 | 18.8|
> |Out-of-domain Only|Code Llama| 76.0| 35.7|
> |Synthetic In-domain Only|Code Llama|78.0 |34.9|
> |ODIS|Code Llama|**78.7** |**40.4**|
>
> [1] Yu, Tao, et al. "Spider: A Large-Scale Human-Labeled Dataset for Complex and Cross-Domain Semantic Parsing and Text-to-SQL Task." Proceedings of the 2018 Conference on Empirical Methods in Natural Language Processing. 2018.

---

### Official Review · Reviewer_TqYu · 2023-08-02

**Soundness:** 3

**Excitement:**

3: Ambivalent: It has merits (e.g., it reports state-of-the-art results, the idea is nice), but there are key weaknesses (e.g., it describes incremental work), and it can significantly benefit from another round of revision. However, I won't object to accepting it if my co-reviewers champion it.

**Paper Topic And Main Contributions:**

This work studies how to select demonstrations for text-to-SQL generation using in-context learning. The authors assume that in-domain examples to hard to obtain and mimic in-domain examples by utilizing retrieved out-domain examples and synthesized in-domain examples. Codex is used for all experiments and improvement is demonstrated on Spider and KaggleDBQA.

**Reasons To Accept:**

The proposed approach demonstrates considerable improvement over baselines.
The paper is easy to understand.

**Reasons To Reject:**

1. The novelty seems limited. Using similarity to retrieve demonstrations is not a new thing.
2. The setting is not justified. Manually annotating 10 ~ 30 in-domain examples for in-context learning seems not a big deal compared to collecting tens of thousands of examples.
3. Codex is a close-sourced deprecated model. I highly encourage the authors to justify the conclusion on open-sourced models like T5 and LLaMA.

**Reproducibility:**

3: Could reproduce the results with some difficulty. The settings of parameters are underspecified or subjectively determined; the training/evaluation data are not widely available.

**Reviewer Confidence:**

4: Quite sure. I tried to check the important points carefully. It's unlikely, though conceivable, that I missed something that should affect my ratings.

---

> ### Author Rebuttal · Authors · 2023-08-29
>
> We appreciate that the reviewer found our work easy to understand and providing considerable improvement over baselines. We also thank the reviewer for their helpful comments and we have made a response to each of their comments.
>
>
> > The novelty seems limited. Using similarity to retrieve demonstrations is not a new thing.
>
> We would like to highlight the novelty and contribution of our work here.
>
> Our work begins with a deep analysis of the effectiveness of gold in-domain demonstrations, where we discovered the SQL distribution within the in-domain data plays a pivotal role. Such analysis was missed in the field of text-to-SQL before our work and we believe our findings would impart valuable insights to future research in the field of text-to-SQL with in-context learning. Furthermore, we were also the first to address the cross-domain text-to-SQL challenge with hybrid sources of demonstrations (out-of-domain examples and synthetic in-domain examples) for in-context learning, in contrast to previous efforts that solely focused on retrieving out-of-domain examples.
>
> Moreover, even retrieving similar examples as demonstration is not a new thing, the specific approach would be largely different in various tasks/settings. For example, synthetically generated in-domain data may not contain similar examples to each specific test example. Therefore, our proposed coverage-based retrieval outperforms traditional similarity-based retrieval by retrieving partially similar examples (shown in Table 4).
>
>
> > The setting is not justified. Manually annotating 10 ~ 30 in-domain examples for in-context learning seems not a big deal compared to collecting tens of thousands of examples.
>
>
> The cross-domain text-to-SQL setting has been a prevalent topic for a long time, encompassing both supervised-learning methods [1-3], and in-context-learning methods [4-6]. Methods in this setting focuses on generalizing to a new database without requiring any data associated with the new database. This setting allows a non-SQL-expert user to create a natural language interface to database (NLIDB) system for any new databases. While annotating 10-30 examples for an SQL expert is not hard, we want to note that annotating 10-30 examples for each new database is extremely costly. For example, the Spider development set has 20 databases. Annotating 10-30 examples for each would require 200-600 SQL annotations. This is the reason that we propose to use pre-existing databases and examples (Spider training set) and synthetically generated examples.
>
> > Codex is a close-sourced deprecated model. I highly encourage the authors to justify the conclusion on open-sourced models like T5 and LLaMA.
>
>
> To demonstrate the generalization of our proposed framework, we also present a study with two strong LLMs: OpenAI GPT3.5 (GPT-3.5-turbo-16k, a.k.a ChatGPT), and Code Llama (CodeLlama-34b-Instruct, the best open-sourced LLMs for code tasks). The results are presented below. Both the out-of-domain demonstration only method and synthetic in-domain demonstration only method outperform the zero-shot setting on two datasets, with GPT-3.5 and Code Llama. Notably, the ODIS approach shows even more improvements in comparison to both the in-domain only and out-of-domain only methods. Due to our limited budget, we did not undertake a hyperparameter search for the experiments with GPT3.5 and Code Llama. Nevertheless, the current results already illustrate the generalizability of our approach.
>
>
> As a result, we believe the remarkable performance of ODIS with both closed-source and open-sourced LLMs on two cross-domain text-to-SQL datasets is able to demonstrate the generalizability of our proposed approach.
>
>
> |Method|LLM|Spider|Kaggle|
> |---|---|---|---|
> |Zero-shot|GPT-3.5|75.7 | 25.7|
> |Out-of-domain Only|GPT-3.5| 80.7| 45.6|
> |Synthetic In-domain Only|GPT-3.5|78.5 | 33.1|
> |ODIS|GPT-3.5|**81.5** |**52.9**|
> ||||
> |Zero-shot|Code Llama|70.8 | 18.8|
> |Out-of-domain Only|Code Llama| 76.0| 35.7|
> |Synthetic In-domain Only|Code Llama|78.0 |34.9|
> |ODIS|Code Llama|**78.7** |**40.4**|
>
>
>
>
>
>
>
> [1] Rubin, Ohad, and Jonathan Berant. "SmBoP: Semi-autoregressive Bottom-up Semantic Parsing." Proceedings of the 2021 Conference of the North American Chapter of the Association for Computational Linguistics: Human Language Technologies. 2021.
>
> [2] Scholak, Torsten, Nathan Schucher, and Dzmitry Bahdanau. "PICARD: Parsing Incrementally for Constrained Auto-Regressive Decoding from Language Models." Proceedings of the 2021 Conference on Empirical Methods in Natural Language Processing. 2021.
>
> [3] Li, Haoyang, et al. "Resdsql: Decoupling schema linking and skeleton parsing for text-to-sql." Proceedings of the AAAI Conference on Artificial Intelligence. Vol. 37. No. 11. 2023.
>
> [4] Poesia, Gabriel, et al. "Synchromesh: Reliable Code Generation from Pre-trained Language Models." International Conference on Learning Representations. 2021.
>
> [5] Li, Jinyang, et al. "Can llm already serve as a database interface? a big bench for large-scale database grounded text-to-sqls." arXiv preprint arXiv:2305.03111 (2023).
>
> [6] Liu, Aiwei, et al. "A comprehensive evaluation of ChatGPT's zero-shot Text-to-SQL capability." arXiv preprint arXiv:2303.13547 (2023).

---

### Official Review · Reviewer_cHQZ · 2023-08-02

**Soundness:** 3

**Excitement:**

3: Ambivalent: It has merits (e.g., it reports state-of-the-art results, the idea is nice), but there are key weaknesses (e.g., it describes incremental work), and it can significantly benefit from another round of revision. However, I won't object to accepting it if my co-reviewers champion it.

**Paper Topic And Main Contributions:**

The paper focuses on the problem of cross-domain text-to-SQL, which translates a natural language question to a SQL query. The authors propose a novel prompt approach that leverages in-domain NLQs and SQLs to improve the performance of codex on this task. The main contributions of the paper are as follows:
1. The authors propose a novel prompt approach that leverages in-domain NLQs and SQLs to improve the performance of LLMs on the cross-domain text-to-SQL task.
2. The authors conduct extensive experiments on two benchmark datasets, Spider and KaggleDBQA, to evaluate the effectiveness of prompts. The results show that new prompts significantly outperforms existing methods and achieves state-of-the-art performance on both datasets.

**Reasons To Accept:**

1. The authors propose a novel prompt approach that leverages in-domain NLQs and SQLs to improve the performance of LLMs on the cross-domain text-to-SQL task.
2. The authors conduct extensive experiments on two benchmark datasets, Spider and KaggleDBQA, to evaluate the effectiveness of prompts. The results show that new prompts significantly outperforms existing methods and achieves state-of-the-art performance on both datasets.

**Reasons To Reject:**

1. The paper lacks experiments on the Spider test set. Without experiments on this dataset, it is difficult to evaluate the generalizability of your proposed approach.
2. While I understand that GPT-4 is expensive, I suggest that you should include experiments with GPT-3.5, which is a more affordable option. This would provide a more comprehensive evaluation of your proposed approach.
3. The examples provided in the appendix are not detailed enough, and it is difficult to find the corresponding prompts for the different methods in Figure 3. I suggest that you provide more detailed examples to help readers better understand your approach.
4. While your proposed approach provides some evidence for in-context learning, I did not find any surprising or novel conclusions in your paper. I suggest that you further develop your approach to provide more significant contributions to the field.

**Reproducibility:**

5: Could easily reproduce the results.

**Reviewer Confidence:**

3: Pretty sure, but there's a chance I missed something. Although I have a good feel for this area in general, I did not carefully check the paper's details, e.g., the math, experimental design, or novelty.

---

> ### Author Rebuttal · Authors · 2023-08-29
>
> We appreciate that the reviewer found our approach novel and effective.  We also thank the reviewer for their helpful comments and we have made a response to each of their comments.
>
>
> > I suggest that you should include experiments with GPT-3.5, which is a more affordable option. This would provide a more comprehensive evaluation of your proposed approach.
>
> We highly value the feedback and agree that experiments on other LLMs would indeed underscore the generalizability of our approach. The table below contains the experimental results with two strong LLMs: OpenAI GPT-3.5 (gpt-3.5-turbo-16k) and an open-sourced model Code Llama [1] (CodeLlama-34b-Instruct). Both the out-of-domain demonstration only method and synthetic in-domain demonstration only method outperform the zero-shot setting on two datasets, with GPT-3.5 and Code Llama. Notably, the ODIS approach shows even more improvements in comparison to both the in-domain only and out-of-domain only methods. Due to our limited budget, we did not undertake a hyperparameter search for the experiments with GPT3.5 and Code Llama. Nevertheless, the current results already illustrate the generalizability of our approach.
>
> As a result, we believe the remarkable performance of ODIS with both closed-source and open-sourced LLMs on two cross-domain text-to-SQL datasets is able to demonstrate the generalizability of our proposed approach.
>
>
> |Method|LLM|Spider|Kaggle|
> |---|---|---|---|
> |Zero-shot|GPT-3.5|75.7 | 25.7|
> |Out-of-domain Only|GPT-3.5| 80.7| 45.6|
> |Synthetic In-domain Only|GPT-3.5|78.5 | 33.1|
> |ODIS|GPT-3.5|**81.5** |**52.9**|
> ||||
> |Zero-shot|Code Llama|70.8 | 18.8|
> |Out-of-domain Only|Code Llama| 76.0| 35.7|
> |Synthetic In-domain Only|Code Llama|78.0 |34.9|
> |ODIS|Code Llama|**78.7** |**40.4**|
>
>
>
> > The paper lacks experiments on the Spider test set. Without experiments on this dataset, it is difficult to evaluate the generalizability of your proposed approach
>
> Our proposed approach involves a preprocessing stage to generate synthetic examples with LLMs which is currently not supported by the Spider evaluation. We are contacting the Spider authors to provide an evaluation on the hidden test set. We would kindly point out that many other studies have also conducted experiments using the Spider development set [2-7], as opposed to the hidden test set. Moreover, it is important to note that the hyperparameters are selected based on a subset of the Spider training set for all experiments on the Spider development set and KaggleDBQA.
> As a result, we believe the remarkable performance of ODIS with three LLMs (Codex, GPT-3.5, and Code Llama) on the Spider development set and the KaggleDBQA dataset is able to demonstrate the generalizability of our proposed approach.
>
>
> > The examples provided in the appendix are not detailed enough, and it is difficult to find the corresponding prompts for the different methods in Figure 3.
>
> We appreciate the feedback and we will add a full prompt example for each experiment in the appendix in the revised version. The code will be released as well.
>
> >  I suggest that you further develop your approach to provide more significant contributions to the field.
>
> We appreciate the feedback. We would like to highlight the novelty and contribution of our work here.
>
> Our work begins with a deep analysis of the effectiveness of gold in-domain demonstrations, where we discovered the SQL distribution within the in-domain data plays a pivotal role. While the insight may not be surprising to the reviewer, such analysis was missed in the field of text-to-SQL before our work and we believe our findings would impart valuable insights to future research in the field of text-to-SQL with in-context learning.
>
> Furthermore, we were also the first to address the cross-domain text-to-SQL challenge with hybrid sources of demonstrations (out-of-domain examples and synthetic in-domain examples) for in-context learning, in contrast to previous efforts that solely focused on retrieving out-of-domain examples. Our empirical findings, substantiated by comprehensive experiments involving both open-sourced and closed-sourced LLMs on the Spider and KaggleDBQA datasets, effectively show the effectiveness of our approach.
>
> Given the remarkable results of our work, we believe it is a good direction to enhance LLMs’ capability on text-to-SQL with the hybrid source of demonstrations, which would stimulate future works on refining data synthesizing methods and retrieval methods.
>
> [1] Rozière, Baptiste, et al. "Code Llama: Open Foundation Models for Code." arXiv preprint arXiv:2308.12950 (2023).
>
> [2] Wang, Lihan, et al. "Proton: Probing schema linking information from pre-trained language models for text-to-sql parsing." Proceedings of the 28th ACM SIGKDD Conference on Knowledge Discovery and Data Mining. 2022.
>
> [3] Ni, Ansong, et al. "Lever: Learning to verify language-to-code generation with execution." International Conference on Machine Learning. PMLR, 2023.
>
> [4] Shaw, Peter, et al. "Compositional Generalization and Natural Language Variation: Can a Semantic Parsing Approach Handle Both?." Proceedings of the 59th Annual Meeting of the Association for Computational Linguistics and the 11th International Joint Conference on Natural Language Processing (Volume 1: Long Papers). 2021.
>
> [5] Qin, Bowen, et al. "SUN: Exploring Intrinsic Uncertainties in Text-to-SQL Parsers." Proceedings of the 29th International Conference on Computational Linguistics. 2022.
>
> [6] Poesia, Gabriel, et al. "Synchromesh: Reliable Code Generation from Pre-trained Language Models." International Conference on Learning Representations. 2021.
>
> [7] Pourreza, Mohammadreza, and Davood Rafiei. "Din-sql: Decomposed in-context learning of text-to-sql with self-correction." arXiv preprint arXiv:2304.11015 (2023).

---

### Meta-Review · Area_Chair_mskR · 2023-09-09

**Recommendation:** 4

**Metareview:**

The paper proposes a framework for selecting demonstration instances for in-context learning of generating SQL queries from text descriptions with LLMs. Unlike existing approaches, which focus on in-domain in-context learning, which requires hand-annotated in-domain examples, the proposed framework ODIS (1) selects the most useful demonstrations from other domains and (2) generates synthetic in-domain demonstrations. The experimental results reported by the authors demonstrate gains over state-of-the-art approaches on two different cross-domain text-to-SQL benchmarks (although the size of the gains is drastically different between the two datasets).

While the retrieval of NLQ(uestion)-to-SQL demonstrations from other domains is already an established practice, the authors show, through a series of soundly executed ablation experiments, that it is the combination of retrieval out-of-domain demonstrations and synthesized in-domain examples that yields optimal performance. The gains over zero-shot NLQ-to-SQL have been reported for three LLMs (Codex in the submission, and then additionally ChatGPT and LLama in the rebuttal), indicating that ODIS is a robust approach, agnostic to the backbone LLM. A particularly valuable aspect of this work, which would be informative for future research efforts in NLQ-to-SQL, is the initial analysis of the distribution of SQL constructs in the in-domain demonstrations and the corresponding effectiveness on in-context learning performance.

While the work is methodologically sound (especially the aforementioned analysis), the extent of novelty of both (1) similarity/coverage-based retrieval of out-of-domain examples as well (2) synthesis of in-domain examples from the NLQs with LLMs, is debatable, and subjective impressions of what is "novel enough" do permeate the reviews. I would, however, hold one thing against the empirical thoroughness of this work: lack of empirical comparison against (a) gold in-domain few-shot in-context learning (as the reviewer TqYu says: what happens if you annotate 10-30 gold in-domain examples) and (b) few-shot fine-tuning (in some parameter-efficient manner, e.g., with QLoRA, with a backbone stronger than the somewhat outdated T5), in addition to only in-context learning. While the authors argue that labeling 10-30 gold examples for 20 or so domains of Spider is "extremely expensive", and that this would not be realistic in the real-world with very many datasets, I'd beg to differ: any developer who wants/expects their database to be extensively queries by non-experts would most likely gladly create more than 30 NLQ-to-SQL examples to facilitate their database/app usage. This is why I'm not entirely sure that zero-shot NLQ-to-SQL with LLMs is the most competitive baseline the authors could have adopted -- if annotating a small number of NLQ-SQL pairs (e.g., 30) surpasses the performance of ODIS, then I'd question that ODIS would be widely used and have a large impact. That said, the analysis part that focuses on identifying in-domain SQLs that are more useful, is unaffected by these concerns. And also, these concerns don't seem to be the issue of this work alone, but the whole body of work on cross-domain transfer for NLQ-to-SQL generation.

Overall, I find this work to be sound and somewhat novel.

---

### Decision · Program_Chairs · 2023-10-07

**Decision:**

Accept-Findings

**Comment:**

The paper proposes a framework for selecting demonstration instances for in-context learning of generating SQL queries from text descriptions with LLMs. Unlike existing approaches, which focus on in-domain in-context learning, which requires hand-annotated in-domain examples, the proposed framework ODIS (1) selects the most useful demonstrations from other domains and (2) generates synthetic in-domain demonstrations. The experimental results reported by the authors demonstrate gains over state-of-the-art approaches on two different cross-domain text-to-SQL benchmarks (although the size of the gains is drastically different between the two datasets).

While the retrieval of NLQ(uestion)-to-SQL demonstrations from other domains is already an established practice, the authors show, through a series of soundly executed ablation experiments, that it is the combination of retrieval out-of-domain demonstrations and synthesized in-domain examples that yields optimal performance. The gains over zero-shot NLQ-to-SQL have been reported for three LLMs (Codex in the submission, and then additionally ChatGPT and LLama in the rebuttal), indicating that ODIS is a robust approach, agnostic to the backbone LLM. A particularly valuable aspect of this work, which would be informative for future research efforts in NLQ-to-SQL, is the initial analysis of the distribution of SQL constructs in the in-domain demonstrations and the corresponding effectiveness on in-context learning performance.

While the work is methodologically sound (especially the aforementioned analysis), the extent of novelty of both (1) similarity/coverage-based retrieval of out-of-domain examples as well (2) synthesis of in-domain examples from the NLQs with LLMs, is debatable, and subjective impressions of what is "novel enough" do permeate the reviews. I would, however, hold one thing against the empirical thoroughness of this work: lack of empirical comparison against (a) gold in-domain few-shot in-context learning (as the reviewer TqYu says: what happens if you annotate 10-30 gold in-domain examples) and (b) few-shot fine-tuning (in some parameter-efficient manner, e.g., with QLoRA, with a backbone stronger than the somewhat outdated T5), in addition to only in-context learning. While the authors argue that labeling 10-30 gold examples for 20 or so domains of Spider is "extremely expensive", and that this would not be realistic in the real-world with very many datasets, I'd beg to differ: any developer who wants/expects their database to be extensively queries by non-experts would most likely gladly create more than 30 NLQ-to-SQL examples to facilitate their database/app usage. This is why I'm not entirely sure that zero-shot NLQ-to-SQL with LLMs is the most competitive baseline the authors could have adopted -- if annotating a small number of NLQ-SQL pairs (e.g., 30) surpasses the performance of ODIS, then I'd question that ODIS would be widely used and have a large impact. That said, the analysis part that focuses on identifying in-domain SQLs that are more useful, is unaffected by these concerns. And also, these concerns don't seem to be the issue of this work alone, but the whole body of work on cross-domain transfer for NLQ-to-SQL generation.

Overall, I find this work to be sound and somewhat novel.